# Parameters of Auditory Evoked Related Potentials P300 in Disorders of Different Cognitive Function Domains (Visuospatial/Executive and Memory) in Elderly Hypertensive Persons

**DOI:** 10.3390/diagnostics13091598

**Published:** 2023-04-30

**Authors:** Liliya Poskotinova, Nina Khasanova, Anna Kharak, Olga Krivonogova, Elena Krivonogova

**Affiliations:** 1N. Laverov Federal Center for Integrated Arctic Research of the Ural Branch of the Russian Academy of Sciences, 163020 Arkhangelsk, Russia; 2Department of Family Medicine and Internal Medicine, Northern State Medical University, 163069 Arkhangelsk, Russia

**Keywords:** hypertension, evoked related potential, cognitive disorders

## Abstract

The neurophysiological correlates of certain types of cognitive impairment in relation to the spatial pattern of auditory cognitive evoked-related potentials (ERPs) in hypertensive persons remain unclear. The aim of this study was to determine the parameters of ERPs (N200, P300) in impaired different domains (visuospatial/executive and memory) of cognitive function in arterial hypertension, including cardiovascular ischemic events. A total of 46 patients (65–84 years) were observed. The clinical diagnosis of vascular dementia, the Montreal Cognitive Assessment Scale (MoCA test) and the spatial pattern of ERPs (N200, P300) were the parameters used to identify three groups: the Control Group without cognitive impairment (n = 13), the group with a leading memory disturbance (Memory Group, n = 20) and the group with a leading visuospatial/executive disturbance (VS/E Group, n = 13). In persons belonging to the Memory Group, N2 latency was prolonged in the central (C3 C4) and right parietal (P4) brain parts; latency of the motor component (P300) may remain similar to that of the ControlGroup. In persons belonging to theVS/E Group, maximal prolonged recognition time (N2), especially in the left central (C3), frontal-midline (Fz), right parietal (P3) and temporal (P4) brain parts, was observed; P300 latency in the central-midline (Cz) and left anterior-temporal (F7) brain parts among all the groups was revealed.

## 1. Introduction

Vascular cognitive impairment includes conditions associated with impaired cerebral blood flow in the brain and subsequent cognitive decline [1]. One of the important conditions inwhich vascular cognitive impairment develops is arterial hypertension (AH). It has been shown that the formation of AH (blood pressure (BP) over 140/90 mmHg) in middle age is a risk factor for cognitive impairment in later years, especially verbal memory, voluntary attention, information processing speed and executive function. However, an increase in systolic BP alone in individuals closer to old age is not always associated with cognitive impairment [2]. Post-stroke dementia has traditionally been thought to be a major contributor to vascular cognitive impairment, but it is now clear that it accounts for only a small proportion of such impairments, with other early neurovascular changes (subcortical, multi-infarct and mixed dementias) making a priority contribution to cognitive impairment in the setting of cardiovascular pathology [3,4].

The Montreal Cognitive Assessment Scale (MoCA test) is widely used to assess cognitive function in individuals with vascular impairment in different parts of the brain, and it is important to determine not only the overall MoCA test score, but also the leading domains of cognitive impairment (executive function, naming, attention, memory and visual-spatial orientation) [5]. However, the difficulty of using only neuropsychological tests (including the MoCA test) to assess early signs of cognitive impairment and dementia, especially in older patients, during cardiac rehabilitation has been noted [6]. The additional use of the cortical long latent cognitive evoked event-relation potentials (ERPs) P300 technique can improve the prognostic value of cognitive impairment in individuals with arterial hypertension [7].

The early ERP components (P100, N100) are related to sensory perception [8]; the N200 component is related to the process of meaningful stimulus recognition and cognitive control, and the N2-P3 (or P300) component reflects the motor component of ERPs in decision-making associated with recognizing a meaningful stimulus [9].

Although the latency of ERPs is significantly greater in older adults with cognitive and mental disorders compared to healthy older adults, it is still difficult to conclude the association of ERP P300 parameters with gender, education and other cognitive tests [10]. The importance of studying the components of the ERP P300 (N100, P200, N200), whose changes correlate with the total score of the MoCA test in elderly persons, is increasing. It is shown that the correlation of latency and amplitude of cortical potentials with cognitive processes reveals direct and medium strength correlations between low scores on the MoCA test and P2 amplitude of ERPs [11]. That is, there is a correlation between long latent brain potentials and cognitive abilities in the elderly, manifested by an increase in P2 of the ERP amplitude and impaired sound decoding [12].

Our previous studies have also shown the significance of studying the latency of the component N2 ofERPs in individuals with arterial hypertension in middle-aged people living in the Arctic zone of the Russian Federation.It has been shown that it is the prolonged latency of the N200component ERPs in the central, frontal, parietal and left temporal brain regions, but not the P300 component, that may be a biomarker of cognitive impairment in middle-aged individuals with increased blood pressure [12]. The neurophysiological correlates of certain types of cognitive impairment, such as memory and visuospatial/executive disturbances as measured by neuropsychological testing, in relation to the spatial pattern of ERPs in hypertensive subjects remain unclear.

We hypothesize that the prediction of cognitive impairment progression by ERP data is more effective for certain leading types of impairment, particularly against a background of cardiac rehabilitation and optimization of cerebral blood flow in hypertensive patients. The basis for this assumption is the advances in the prognostic value of ERPs during transcranial magnetic stimulation of precisely selected cortical areas of the brain indiagnosing specific cognitive impairments, such as those associated with impaired visual and spatial functions after a stroke [13]. The aim of the study was to determine the parameters of auditory cognitive ERP P300 in impaired different domains (visuospatial/executive and memory) of cognitive function in hypertensive individuals.

## 2. Materials and Methods

A non-randomized controlled trial was carried out in 2022 (May–June) at the Arkhangelsk Veterans Hospital (Russian Federation). All patients gave written informed consent to participate in the study. This study was conducted in accordance with the Declaration of Helsinki and approved by the Ethics Committee of N. Laverov Federal Centre for Integrated Arctic Research of the Ural Branch of the Russian Academy of Sciences (Protocol Code 4, 10 February2022) for studies involving humans. 

The sample included 46 patients (39 women and 7 men) who had received treatment for the underlying disease of arterial hypertension (14 days inpatient treatment). Among the patients, 60% had higher professional education and 40% had secondary professional education. Participants had a body mass index of 18.5 to 34.2 kg/m^2^ (no more than 30% of those with obesity class I), which indicated no significant effect of metabolic impairment on cognitive function (obesity-associated cognitive impairment) [14].

Criteria for inclusion in the main group:(1)age 65–84 years (young-old and middle-old);(2)normal state of consciousness and ability to answer questions;(3)anamnesis of arterial hypertension (BP more than 140/90 mmHg) [15];(4)NINDS-AIREN (National Institute of Neurological Disorders and Stroke and the Association Internationale pour la Recherche et l’Enseignementen Neurosciences) criteria for vascular dementia [16]:
The presence of a syndrome of cognitive impairment (impairment of goal formation, abstraction, initiation, planning, organization and maintenance of activities; memory impairment: impaired reproduction with relatively preserved recognition and effectiveness of cues);Presence of cerebrovascular disease on magnetic resonance imaging data: marked hypointense irregular “patchy” foci located periventricularly and in deep white matter or diffuse low-density symmetrical changes in the semiovascularcentre projection in association with at least one lacunar foci; presence of focal symptoms in the neurological status—hemiparesis, weakness of the lower part of the facial muscles, Babinski’s symptom, sensory disturbances, dysarthria, walking disorders, extrapyramidal symptoms, which can be explained by the presence of focal subcortical localization;Presence of a temporal relationship between dementia and cerebrovascular disorders: stepwise progression of cognitive impairment;(5)MoCA score of less than 26.

The main group included 6 people out of 33 (18.2%) who had suffered an acute cerebrovascular event (CVE—ischemic stroke, transient ischemic attack) with a duration of 1 to 6 months (n = 1) and more than 6 months (n = 5) by the time of the study. There were three of them with mild hemiparesis of the arm or leg. Transient ischemic attacks were all in the vertebro-basilar vascular system and ischemic strokes were in the vertebro-basilar system (n = 1) and left (n = 1) and right carotid vascular areas (n = 3).

Criteria for inclusion in the control group:(1)age 65–84 years;(2)normal state of consciousness and ability to answer questions;(3)anamnesis of arterial hypertension (BP more than 140/90 mmHg) [15];(4)no clinical signs consistent with vascular dementia according to the NINDS-AIREN criteria;(5)MoCA test score of 26 and higher.

The control group included 8 of 13 (61.5%) individuals who had suffered CVE (ischemic stroke, transient ischemic attack) with a time course of 1 to 6 months (n = 3) and more than 6 months (n = 5) by the time of the study, without motor impairment at the time of the study.The transient ischemic attacks were in the vertebro-basilar system and the ischemic strokes were in the vertebro-basilar system (n = 1) and left (n = 1) and right (n = 1) carotid areas.

Non-inclusion criteria for both control and main groups (n = 46):(1)neurosensory hearing loss above grade I (sound perception lower than 25 dB);(2)Barthel Index score of 60 or lower (severe or total dependency in daily life);(3)Psychiatricdisorders, severe dementia;(4)decompensation of comorbid conditions;(5)severe motor and sensory aphasia.

MoCA test was used to assess eight cognitive domains including visuospatial/executive function, naming, memory, attention, language, abstraction, delayed recall and time-space orientation. A score of 26–30 corresponded to no cognitive impairment, while a score of 25 or lower corresponded to cognitive impairment [17]. 

As a result, the main group with cognitive impairment (n = 33, 29 women and 5 men) and the control group without verified cognitive impairment (n = 13, 11 women and 2 men) were formed. Using the MoCA test, the leading impairments of the cognitive domains were identified: visuospatial/executive impairments (n = 13), memory and speech impairments (n = 2) and memory impairments (n = 18).

Subsequently, three groups were considered: the Control Group (n = 13), the group with leading visuospatial/executive cognitive impairment (n = 13, VS/E Group) andthe group with leading memory impairment (Memory Group, n = 20).

Study participants from all groups received combination anti-hypertensive therapy (Table 1). Five participantsdidnot continuously take anti-hypertensive medication during the study due to episodes of low blood pressure. Complementary therapy included anti-platelet/anti-coagulant medicine and neurometabolic medicine.

At the end of the course of the therapy, the hemodynamic parameters were stabilized and the general condition of the patients improved. Systolic and diastolic BP (mmHg) and heart rate (HR, bpm) were measured using the upper arm blood pressure monitor UA-654MR (A&D, Asahi, Kitamoto-shi, Saitama, Japan). This study presents data on BP and HR at the end of the treatment when the general condition and hemodynamic parameters had stabilized.

At the end of hospital treatment, all patients were assessed for auditory cognitive ERPs parameters using the Neuron-Spectrum-4/VPM electroencephalograph («Neurosoft», St.Petersburg, Russia). These studies were carried out at the medical company «BrainPro» (Arkhangelsk, company director N. Sydorov) with the support of theneuropsychologist E. Sydorova. Participants were seated on an armchair in a sitting position. 

The scalp was cleaned and then electrodes were fixed on the scalp surface using high conductivity gel at the electroencephalogram points in the 10–20 system (F3 F4 F7 F8 C3 C4 P3 P4 T3 T4 FzPzCz). Two auricular reference electrodes, A2 (right) and A1 (left), and electrode FPz (ground) were used in the wiring scheme. The impedance between the electrodes was kept below 5 kΩ. The oddball paradigm was used. Binaural non-verbal acoustic stimulation conditions were: stimulus duration 50 ms, intensity 80 dB, period between stimuli 1 s, tone 2000 Hz with 25% occurrence of a significant stimulus, tone 1000 Hz with 75% occurrence of an insignificant stimulus;twenty-fiveaverages of values for the significant stimulus were used, and P300 latency (motor component by press button) was defined as the maximum positive component with a latency of 250–500 ms with N2-P3 amplitude (µV). N200 latency (ms) and P2-N2 amplitude (µV) were determined.

Statistical processing of the materials was carried out using the Statistica 10.0 program (USA, Tulsa, Oklahoma, StatSoft). Due to the fact that the vast majority of the studied parameters were not normally distributed, they are described using the median (Me) and 25th and 75th percentiles (Me (25p;75p)). Comparison of quantitative variables between several independent groups was carried out using the multiply Kruskal-Wallis H criterion. Then, in order to clarify which groups showed differences, a pairwise comparison using the Mann-Whitney U test was used, with the correction of the significance criterion for multiple comparisons (for three groups, the critical significance level was *p* < 0.017). In correlation analysis, the non-parametric Spearman’s coefficient (*p* < 0.05) was used.

## 3. Results

The results of the analysis of the anamnestic and objective data of the study participants are presented in Table 2. Participants were statistically similar in age, duration of arterial hypertension and hemodynamic parameters at the time of the study; MoCA test score values in the Memory Group and VS/E Group were statistically identical, but significantly lower than those in the Control Group. In the time since the acute CVE, patients in all groups mostly corresponded to a late recovery period (more than 6 months after the event). In terms of localization of the acute CVE, the groups also did not have significant priorities in lateralization (there were both left- and right-hemispheric vascular events). 

Thus, the groups were comparable in terms of the incidence of CVE in different brain areas (involving the right and left brain hemispheres). Only three participants out of 33 (9%) had residual mild hemiparesis in the groups with cognitive impairment. Preliminary analysis showed that there was no significant inter-hemispheric asymmetry of the ERP parameters in these individuals, so they remained included in the groups with cognitive impairment.

Individuals in the VS/E Group had the highest latency of the N200 component of ERPs, especially in the frontal, central, parietal and right temporal regions (Table 3).

The latency of the N200 component in the Memory Group was higher than that in the Control Group, especially in the central (C3 C4) and right parietal (P4) brain parts. Compared to the Control Group, individuals in the VS/E Group had the highest N200 latency, particularly in the frontal (F3 F4), central (C3 C4), right parietal (P4) and temporal (T4), and midline (FzCz and Pz) brain regions. Further, N200 latency was significantly higher in the VS/E Group compared to that in the Memory Group, especially in the left central (C3), right temporal (T4) and frontal midline(Fz) brain regions. 

N200 amplitudes varied widely from 1 to 17 µV but median values were less than 3–5 µV, so no significant patterns between the groups were observed.

The latency of the P300 component of ERPs, which reflects the speed of decision-making under conditions of detecting a significant auditory stimulus, was maximal in the VS/E Group, especially in the midline (FzCz) and left anterior temporal (F7) brain parts (Kruskal-Wallis H test) (Table 4). In a pairwise comparison of the three groups (Mann-Whitney U test), the P3 latency in the left anterior temporal region (F7) was highest in the VS/E Group.

The amplitudes of the P300 component of ERPs had a large variability, so their mean values were not statistically different between the groups (Table 5).

Examples of comparisons of the N200 and P300 component latencies in their cerebral distribution in three participants from different groups are shown in Figure 1. 

The participant in the Memory Group(Figure 1b) had a slightly different N200 latency than the participant in the Control Group (Figure 1a), but with the presence of right-sided asymmetry in the anterior temporal brain parts (F7 F8). The largest N200component latency compared to participants in other groups was reflected in the VS/E Group participant (Figure 1c), especially in the left and right frontal (F3 F4) and anterior temporal brain regions (F7 and F8). This VS/E Group participant also had prolonged latency of the P300 motor component, especially with left-sided asymmetry in frontal areas (F3 F4) and right-sided asymmetry in anterior temporal areas (F7 F8). This example demonstrates the presence of different variants of the cerebral distribution of interhemispheric asymmetries of the latencies of both N200 and P300components, highlighting individual differences in the neurophysiological mechanisms of cognitive impairment formation in individuals with cardiovascular pathology.

Correlation analysis of MoCA test data and ERP parameters revealed significant correlations (*p* < 0.05) only with N2 component latency in individuals of the VS/E Group (Table 6). 

The results indicate a correlation of cognitive decline with increased N200 latency, primarily in the frontal midline (Fz) and right parietal (P4) and temporal (T4) brain parts in patients with visuospatial/executive cognitive impairment.

## 4. Discussion

The results showed that the same level of decline in the MoCA test score may reflect different electro-neurophysiological mechanisms of cognitive impairment in individuals with prolonged arterial hypertension. In individuals with a leading memory impairment, motor response time (P300 component latency) may remain at the Control Group level, while recognition time may increase only in certain brain areas, in this case, the central and right parietal regions of the brain. Individuals with leading VS/E cognitive impairments, despite identical levels according to the MoCA test to the group with memory impairments, showed the maximum prolonged significant stimulus recognition time in almost all EEG leads. However, the greatest dynamics of N200latency lengthening from the Control Group to the VS/E Group was expressed in the left central (C3), frontal-midline (Fz) and right parietal and temporal brain parts (P4 T4). In patients with VS/E impairments, correlation analysis suggests that cognitive decline is most strongly associated with longer cognitive control processing times in the frontal midline (Fz) and right parietal and temporal brain regions (P4 T4). The latency of the motor component P300 did not show such significant changes in the groups, with the exception of lengthened time in the central midline (Cz) and left anterior temporal brain (F7) in those with VS/E impairments. In contrast, the latency of P300 in individuals with memory impairment tends to decrease compared to that in the Control Group, which requires a separate study.

Apparently, medicines that improve brain microcirculation, especially anti-hypertensive medicines, optimize working memory processes in the first place. For example, prescribing valsartan (angiotensin-converting enzyme inhibitor) to elderly hypertensive patients has been shown to provide a significant reduction in P300 latency 6 months after antihypertensive treatment. At the same time, according to neuropsychological testing, word list recall, word list recall test, word list recognition, i.e., cognitive functions of the memory domain improved [18]. There are studies demonstrating the relationship between cognitive decline (executive function, praxis, language, attention and memory) and prolongation of P300 latency in the central midline and left central brain parts in hypertensive elderly individuals in the acute period after ischemic stroke (up to 4 weeks) [19]. It is believed that in the later recovery period after antihypertensive and neurometabolic treatment, vascular brain disorders (cerebral microangiopathies), especially in the periventricular white matter of the posterior parts of the left frontal lobe, the middle part of the right cingulate gyrus and the posterior-medial part of the cerebellar body, persist in those with VS/E cognitive impairment [20].

The priority of temporal brain areas in the lengthening of both recognition and motor components of the ERPs may be related not only to vascular abnormalities, but also to the accession of neurodegenerative disorders. In our previous studies, it was shown that the severity of signs of such a neurodegenerative disease as Parkinson’s disease correlates with a decrease in ERP P300 parameters in temporal and especially anterior temporal regions [21]. It has also been noted that in Alzheimer’s disease, the stronger cerebral ERP P300 generators often shift from the frontal lobes to the temporal regions. The decrease in the strength of the lower frontal source in P300 generation and the shifting of the zone of maximum intensity to the parietal and upper temporal regions of the brain suggest that these areas are of particular importance in the study of neurodegenerative diseases [22].

In transient ischemic attacks in the vertebro-basilar system, there may also be cognitive effects of vestibular insufficiency, which may be associated with decreased visuospatial and executive function. There is a decrease in functional connectivity between the cortical Brodmann fields (right wedge-shaped area and left parietal gyrus), which may be ‘early biomarkers’ of visuospatial, executive functions and attention hypofunction. These features have been associated with decreased P300 amplitude in these areas [23], but this pattern was not present in our study due to the various CVEs in the groups.

Another reason for priority elongation of recognitionprocessing times in persons with VS/E impairments is the possible presence of neglect syndrome following brain damage, especially in the elderly. Visual-spatial attention disorder, visual sensory (input) neglect, may be a cause of difficulties in performing MoCA tests on certain tasks (drawing clocks, geometric shapes, including with a sequence of letters and numbers). The syndrome of visuospatial neglect is one of the phenomena of optic-spatial agnosia, which occurs when there is unilateral damage to the cortical structures of the posterior brain regions and when visual function is preserved, but the neural networks underlying this recovery are not well understood [24,25]. Further, temporarily occurring and transient visual neglect may be part of a cognitive-affective syndrome following cerebellar circulatory disturbances, which is very difficult to suspect either clinically or by paper-based testing [26].

Recovery of reduced visual-spatial attention function after vascular cerebral disturbances in the right hemisphere in the early recovery period (4 weeks after the event) was associated with bilateral activation of cerebral areas. However, the persistence of reduced visual-spatial attention function after suffering cerebral ischemia was found predominantly in the right cortical areas—superior and medial frontal, inferior temporal, medial orbitofrontal, inferior parietal gyrus, right inferior temporal gyrus, right medial orbitofrontal cortex and superior temporal pole in the right hemisphere [25].

Authors Lin-Lin Ye et al. [27] also considered groups with and without visual-constructive impairment when analyzing the parameters of sensory components (P100, N100) and the motor component of ERP P300 after a right hemisphere brain stroke. In both the group with and without visual-constructive impairments, P300 latency was increased compared to that in the control group (without stroke), but not between the groups with and without visual-spatial impairments. The latency of the sensory component of P1 was longer and the amplitude of N100 was lower in the frontal regions in the group with cognitiveimpairments. The authors concluded that visual-spatial attention function is impaired after stroke in the right hemisphere, as reflected in subclinical signs of visual-spatial impairment [27]. Probably, the amplitudes of the early sensory component (P100, N100) of ERPs are more sensitive to visual-spatial impairments than N200, since according to our data, the amplitudes of the more later component of the ERPs (N200) did not differ in the groups.

It is important to note that we obtained data on the prolongation N200latency not only in patients who had suffered acute CVE, but also in those without verification of these events. It can be assumed that patients in our study may have suffered so-called «silent» strokes [28], including involving the right hemisphere, which may have affected both the reduction in gray matter volume in the left upper and lower frontal gyrus, left upper and right temporal gyrus and bilateral parahippocampal gyri. There is a correlation between decreased volume of frontal, temporal and hippocampal areas and prolongation of P300 latency [28]. The present study did not aim to clinically identify the neuropsychological syndrome, which requires a specific instrument for neuropsychological testing. However, it is the combined decline in cognitive function on the MoCA test with a leading domain of visual-constructive impairment and a marked lengthening of the N200 component in certain brain regions, especially with interhemispheric asymmetry, that may help suspect the presence of a neoplegic syndrome and refer this patient for additional neurological examination and specific neuropsychological testing.

Bottiroli S. et al. (2017) used the MoCA test subscales (executive functions, working memory, visual-spatial processing, language and orientation) as effectiveness criteria for cognitive skills training in elderly people in the 3D virtual reality platform Smart Aging [29].The use of data from the MoCA subscales allowed the authors to conclude that this virtual platform appears to be a good tool for assessing primarily executive functions, working memory and visual-spatial processes [29]. A double-blinded randomized control trial of multidomain attention training for alertness, sustained attention and visual-spatial attention in older adults with mild cognitive impairment (MCI) was performed [30]. Some of the indicators of the effectiveness of such training were the subscales of the MoCA test (sustained attention and visual-spatial attention). Significant improvements in attention, memory and orientation on the values of the MMSE and MoCA subscales and better results compared to the control group 6 months after performing multidomain cognitive training have been shown [30]. Thus, the data we obtained on the spatial cerebral distribution of N200 and P300 components of ERPs in individuals with leading cognitive impairments by the MoCA test (in this case, memory and visual/spatial and executive functions) can be used as clarifying neurophysiological biomarkers for selecting persons for various kinds of cognitive training in older adults. Although, specialized neurophysiological tests can additionally provide more exact information about operative memory processes and visual/spatial and executive functions in these cases.

Comorbid autoimmune conditions such as autoimmune Hashimoto’s thyroiditis [31] or Sjögren’s disease [32] contribute to cerebral mechanisms of prolongation of significant stimulus recognition time (N200) and decision time (P300), especially in older persons. Although the present study did not include people with the above pathology in the sample, a more thorough examination of patients is needed in the future because systemic autoimmune diseases may also influence the results of ERPs.

Identification of visual cognitive evoked-related potentials will be useful in the study of leading visual-spatial disorders developing in hypertensive individuals. The combined study of the spatial brain distribution of ERPs with different afferent inputs (both auditory and visual) in persons with CVE (stroke, transient ischemic attack) may help to identify the most significant brain areas that can be used for targeted bioelectric and magnetic treatment, for example, during transcanial magnetic stimulation.

The limitations of the study may be related to the limited number of subjects, as well asto the inclusion of individuals with different lateralization of cerebral circulation disorders (both right- and left-hemispheric), although studying the relationship between ERP parameters and vascular disorders in strokes of different localization seems promising in the next phase of the study. The predominance of females in the sample made it impossible to distinguish sex-specific features of ERP parameters, although in the elderly age, the sex differences may not be as significant as in middle age. In terms of time after CVE, patients in both the control and the main group mostly corresponded to a late recovery period (more than 6 months after the event), but in recruiting large samples it would be better in the future to separate the neurophysiological findings by periods up to 6 months and more than 6 months after the cardiovascular disturbances.

## 5. Conclusions

In hypertensive persons with a leading cognitive memory impairment, motor response time according to ERPs (P300 latency) may remain similar to control group values without cognitive impairment, while recognition time (N200latency) is prolonged in the central (C3 C4) and right parietal (P4) brain regions. Hypertensive persons with leading visuospatial/executive impairments, despite identical levels on the MoCA test to the group with leading memory impairments, showed maximal prolonged recognition time of a significant auditory stimulus (N200), especially in the left central (C3), frontal-midline (Fz) and right parietal and temporal brain parts (P4 T4). The latency of the motor ERP component (P300) in the group with leading visuospatial/executive impairments was maximal in the central-midline (Cz) and left anterior-temporal (F7) brain parts.

## Figures and Tables

**Figure 1 diagnostics-13-01598-f001:**
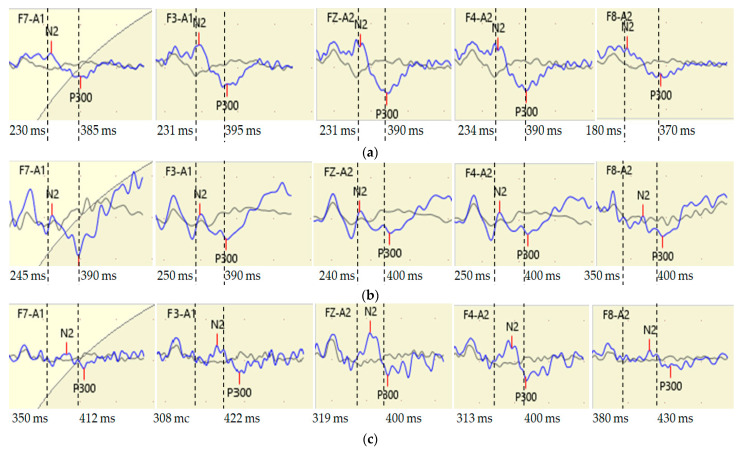
Fragments of the spatial distribution of ERP N200 and P300 components in three participants from the Control Group (**a**), Memory Group (**b**) and VS/E Group (**c**). The dotted lines indicate the latencies of the N2 (N200) and P300 components in the participant of the Control Group (**a**) and further for comparison in the participants of the Memory Group (**b**) and VS/E Group (**c**). Value from the dotted line to the left: N2 latency; value from dashed line to the right: P300 latency.

**Table 1 diagnostics-13-01598-t001:** Distribution of medical treatmentsin the groups ^a^.

Medical Treatments	Control Group	Memory Group	VS/E Group
Anti-hypertensive therapy			
Angiotensin-converting enzyme inhibitors, diuretics andselective beta1 adrenergic receptor blockers	4/30.7	6/30.0	3/23.1
Angiotensin II receptor blockers, diureticsand selective beta1 adrenergic receptor blockers	3/23.1	6/30.1	3/23.1
Angiotensin-converting enzyme inhibitors and calcium channel blockers	1/7.7	1/5.0	1/7.7
Angiotensin II receptor blockersand calcium channel blockers	1/7.7	1/5.0	2/15.4
Angiotensin-converting enzyme inhibitors, calcium channel blockers and selective beta1 adrenergic receptor blockers	2/15.4	4/20.0	2/15.4
No anti-hypertensive treatment during this study ^b^	2/15.4	2/20.0	2/15.4
Total	13/100	20/100	13/100
Complementary therapy			
Anti-platelet /anti-coagulant medicine and 3-hydroxy-3-methylglutaryl coenzyme A reductase inhibitor (statins)	9/69.2	10/50.0	7/53.8
3-hydroxy-3-methylglutaryl coenzyme A reductase inhibitor (statins)	2/15.4	5/25.0	3/23.1
Antidiabetic medicine	2/15.4	5/25.0	3/23.1
Total	13/100	20/100	13/100
Neurometabolic therapy			
Total	13/100	20/100	13/100

^a^ Data are presented as number of persons/percentage; ^b^ These participantsdidnot continuously take anti-hypertensive medication during the study due to episodes of low blood pressure.

**Table 2 diagnostics-13-01598-t002:** Anamnestic data and cardiovascular and cognitive parameters in hypertensive persons with different leading domains of cognitive impairment (Me (25p;75p)) ^a^.

Parameter	Control Group n = 13 (1)	Memory Group n = 20 (2)	VS/E Group n = 13 (3)	*p* Kruskal- Wallis H
Age, years	72.0 (65.0; 74.5)	73.0 (68.0; 76.0)	78.0 ^b (1−2, 2−3, 1−3)^ (73.0; 81.0)	>0.05
Duration of hypertension, years	17.5 (5.5; 32.5)	18.0 (10.0; 31.0)	20.0 ^b (1−2, 2−3, 1−3)^ (5.0; 30.0)	>0.05
Systolic BP, mmHg	130.0 (120.0; 130.0)	132.0 (130.0; 154.0)	140.0 ^b (1−2, 2−3, 1−3)^ (130.0; 140.0)	>0.05
Diastolic BP, mmHg	80.0 (75.0; 80.0)	80.0 (80.0; 86.0)	80.0 ^b (1−2, 2−3, 1−3)^ (80.0; 90.0)	>0.05
HR, bpm	74.0 (65.5; 76.0)	70.0 (65.0; 76.0)	72.0 ^b (1−2, 2−3, 1−3)^ (63.0; 80.0)	>0.05
MoCA test, score	26.0 (26.0; 27.0)	23.0 (20.0; 25.0)	23.0 ^b (2−3), c (1−2, 1−3)^ (19.0; 24.0)	0.001

BP: blood pressure; HR: heart rate; MoCA: Montreal Cognitive Assessment.^a^ Data are presented as median (lower and upper quartiles). ^b^ Statistically significant difference between the groups: *p* > 0.05, the Mann-Whitney U test. ^c^ Statistically significant difference between the groups: *p* < 0.001, the Mann-Whitney U test.

**Table 3 diagnostics-13-01598-t003:** N200 component latency (ms) of ERPs in hypertensive individuals with different leading domains of cognitive impairment (Me (25p;75p)) ^a^.

EEG Leads	Control Group n = 13 (1)	Memory Group n = 20 (2)	VS/E Group n = 13 (3)	*p* Kruskal- Wallis H
F3	221.9 (190.0; 241.1)	246.6 (228.0; 255.0)	306.8 ^b (1−2, 2−3); c (1−3)^ (260.0; 356.0)	0.003
F4	227.4 (194.0; 235.6)	235.0 (191.8; 254.8)	257.8 ^b (1−2, 2−3); c (1−3)^ (252.5; 331.5)	0.007
C3	219.2 (190.0; 238.4)	246.9 (228.0; 255.0)	304.1 ^c (1−2, 2−3, 1−3)^ (255.0; 356.2)	<0.001
C4	205.5 (186.3; 239.0)	254.8 (218.0; 255.0)	260.5 ^b (2−3); c (1−2, 1−3)^ (238.7; 312.3)	*p* = 0.003
P3	198.5 (148.6; 239.8)	234.0 (220.0; 250.0)	312.3 ^b (1−2, 2−3); c (1−3)^ (238.3; 343.0)	*p* = 0.011
P4	213.7 (155.0; 241.5)	239.0 (225.0; 260.0)	268.3 ^b (2−3); c (1−2, 1−3)^ (244.5; 339.7)	*p* = 0.005
F7	213.7 (142.0; 239.0)	239.8 (213.7; 266.0)	220.5 ^b (1−2, 2−3, 1−3)^ (152.0; 300.0)	*p* > 0.05
F8	197.3 (140.0; 238.3)	231.0 (203.0; 250.0)	247.0 ^b (1−2, 2−3, 1−3)^ (137.5; 259.5)	*p* > 0.05
T3	194.0 (167.1; 230.2)	240.0 (214.5; 247.5)	246.6 ^b (1−2, 2−3, 1−3)^ (202.7; 356.0)	*p* > 0.05
T4	167.1 (130.0; 227.4)	220.0 (21.7; 244.0)	266.9 ^b (1−2); c (2−3, 1−3)^ (235.0; 301.4)	*p* = 0.002
Fz	213.7 (172.6; 246.6)	223.0 (219.2; 241.7)	273.9 ^b (1−2); c (2−3, 1−3)^ (250.0; 323.3)	*p* = 0.002
Cz	219.2 (180.8; 250.0)	227.7 (212.9; 250.7)	296.2 ^b (1−2, 2−3); c (1−3)^ (250.0; 334.2)	*p* = 0.014
Pz	219.2 (192.9; 240.1)	231.0 (199.9; 256.15)	263.0 ^b (1−2, 2−3); d (1−3)^ (244.0; 345.2)	*p* = 0.010

^a^ Data are presented as median (lower and upper quartiles); ^b^ Statistically significant difference between the groups: *p* > 0.05, the Mann-Whitney U test; ^c^ Statistically significant difference between the groups: *p* < 0.01, the Mann-Whitney U test; ^d^ Statistically significant difference between the groups: *p* < 0.001, the Mann-Whitney U test.

**Table 4 diagnostics-13-01598-t004:** P300 component latency (ms) of ERPs in hypertensive individuals with different leading domains of cognitive impairment (Me (25p;75p)) ^a^.

EEG Leads	Control Group n = 13 (1)	Memory Group n = 20 (2)	VS/E Group n = 13 (3)	*p* Kruskal- Wallis H
F3	369.8 (336.0; 427.4)	336.0 (300.0; 393.0)	400.02 ^b (1−2, 2−3, 1−3)^ (343.0; 484.9)	>0.05
F4	353.4 (308.0; 413.7)	329.0 (316.0; 404.0)	385.0 ^b (1−2, 2−3, 1−3)^ (335.0; 446.6)	>0.05
C3	386.3 (334.2; 430.1)	346.0 (286.0; 390.0)	380.8 ^b (1−2, 2−3, 1−3)^ (340.0; 457.5)	>0.05
C4	358.9 (300.0; 419.2)	330.0 (318.0; 398.0)	413.3 ^b (1−2, 2−3, 1−3)^ (346.5; 457.5)	>0.05
P3	410.1 (350.7; 432.9)	338.0 (324.0; 386.0)	386.3 ^b (1−2, 2−3, 1−3)^ (302.0; 441.1)	>0.05
P4	361.7 (313.0; 431.5)	332.0 (320.0; 369.0)	401.8 ^b (1−2, 2−3, 1−3)^ (333.5; 454.8)	>0.05
F7	375.3 (336.0; 424.6)	322.0 (296.0; 351.0)	421.9 ^b (1−2, 1−3); d (2−3)^ (348.0; 473.9)	0.006
F8	346.0 (328.0; 391.8)	346.7 (310.0; 393.9)	357.0 ^b (1−2, 2−3, 1−3)^ (324.0; 430.1)	>0.05
T3	385.0 (340.0; 430.1)	329.0 (302.0; 351.0)	388.8 ^b (1−2, 2−3, 1−3)^ (330.0; 446.6)	>0.05
T4	342.0 (316.0; 399.9)	336.0 (313.0; 358.0)	370.9 ^b (1−2, 2−3, 1−3)^ (334.0; 420.5)	>0.05
Fz	367.1 (340.0; 413.7)	319.5 (308.0; 337.9)	419.2 ^b (1−2, 1−3); c (2−3)^ (369.8; 468.5)	0.034
Cz	391.8 (338.0; 424.6)	313.7 (285.0; 349.2)	439.7 ^b (1−2, 1−3); c (2−3)^ (342.0; 517.8)	0.039
Pz	375.3 (349.5; 420.8)	325.0 (307.0; 348.1)	430.1 ^b (1−2, 2−3, 1−3)^ (332.0; 517.8)	>0.05

^a^ Data are presented as median (lower and upper quartiles); ^b^ Statistically significant difference between the groups: *p* > 0.05, the Mann-Whitney U test; ^c^ Statistically significant difference between the groups: *p* < 0.05, the Mann-Whitney U test; ^d^ Statistically significant difference between the groups: *p* < 0.01, the Mann-Whitney U test.

**Table 5 diagnostics-13-01598-t005:** P300 component amplitude (µV) of ERPs in hypertensive individuals with different leading domains of cognitive impairment (Me (25p;75p)) ^a^.

EEG Leads	Control Group n = 13 (1)	Memory Group n = 20 (2)	VS/E Group n = 13 (3)	*p* Kruskal- Wallis H
F3	10.3 (5.4; 12.7)	7.4 (5.3; 12.2)	8.3 (6.8; 14.8)	>0.05
F4	9.13 (7.0; 11.6)	8.4 (5.5; 14.8)	12.0 (5.9; 16.9)	>0.05
C3	9.4 (8.5; 12.1)	8.4 (5.6; 13.6)	7.2 (5.6; 10.2)	>0.05
C4	7.0 (6.3; 10.4)	7.3 (4.0; 13.1)	8.4 (5.5; 15.3)	>0.05
P3	7.8 (3.4; 11.1)	7.0 (2.7; 10.7)	7.8 (6.1; 10.0)	>0.05
P4	5.3 (7.8; 9.15)	7.2 (3.1; 10.8)	8.3 (3.1; 16.5)	>0.05
F7	5.15 (3.6; 7.9)	3.8 (3.0; 7.6)	7.6 (6.9; 8.6)	>0.05
F8	4.9 (2.3; 8.7)	5.1 (3.9; 6.4)	6.6 (2.4; 13.8)	>0.05
T3	5.5 (3.1; 8.2)	5.8 (3.0; 6.6)	7.3 (5.5; 9.0)	>0.05
T4	4.0 (3.1; 7.6)	4.8 (4.0; 7.0)	5.1 (2.1; 11.1)	>0.05
Fz	10.3 (8.0; 12.7)	7.0 (4.4; 10.6)	13.0 (9.2; 18.0)	>0.05
Cz	9.3 (6.0; 12.4)	5.9 (4.3; 9.4)	10.2 (4.7; 10.8)	>0.05
Pz	7.7 (3.6; 10.0)	4.5 (1.6; 10.2)	9.1 (3.9; 20.0)	>0.05

^a^ Data are presented as median (lower and upper quartiles).

**Table 6 diagnostics-13-01598-t006:** Correlation Spearmen coefficients (*p* < 0.05) of N200 component latency and MoCA test values in hypertensive individuals of VS/E Group.

EEG Lead	Spearmen Coefficients	*p*Spearmen
P4	−0.77	0.013
T4	−0.87	0.004
Fz	−0.82	0.007

## Data Availability

Not applicable.

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
