# Peer review of "Parameters of Auditory Evoked Related Potentials P300 in Disorders of Different Cognitive Function Domains (Visuospatial/Executive and Memory) in Elderly Hypertensive Persons"

_diagnostics, 2023, doi:10.3390/diagnostics13091598_

Round 1
Reviewer 1 Report
The work, unfortunately, does not bring any new knowledge on the subject. ERPs have been used for many years in the diagnosis of possible cognitive disorders and they are non-specific potentials, i.e., latency prolongations can occur in various disorders - cognitive, toxic, demyelinating diseases but also in endocrine or rheumatic diseases. The study group is so small that the statistics are unreliable and the evaluation of cognitive function with the MoCa test is nonexistent.
Author Response
Reviewer 1. The work, unfortunately, does not bring any new knowledge on the subject. ERPs have been used for many years in the diagnosis of possible cognitive disorders and they are non-specific potentials, i.e., latency prolongations can occur in various disorders - cognitive, toxic, demyelinating diseases but also in endocrine or rheumatic diseases.
Response. We regret that Reviewer 1 did not find new and meaningful findings in our article.
The development of cognitive disorders occurs due to different pathophysiological mechanisms of brain dysfunctions. Consequently, a detailed study of the ERPs components (sensory, motor, ets) makes it possible to assess the difference in the development of those disorders of brain functions, taking into account the pathogenesis. For example, there is convincing evidence that ERPs parameters differ in the development of neurodegenerative (Alzheimer's disease) and vascular (vascular dementia) cognitive disorders (DOI: 10.3233/JAD-200885).
There is also evidence for variability in the topology of the distribution of the N2 component associated with certain types of cognitive tasks performed (10.1097/WNR.0b013e32834bbe1f).
Undoubtedly, there are reasons to be skeptical about the use of EEG and ERPs in clinical studies in severe vascular cognitive impairment. However, a group of experts notes that ERPs indicators can be considered as biomarkers of "neuronal synchronization" to elucidate the relationship between features of cerebrovascular damages associated with vascular cognitive impairment (doi: 10.1016/j.neurobiolaging).
In our study, we did not set a separate aim to diagnose cognitive disorders according to ERPs, because they have already been diagnosed according to approved clinical criteria, including neuropsychological testing. Our research aims to clarify the prognosis of the development of certain cognitive disorders (in this case, memory impairments and visual-executive functions) according to ERPs data in individuals with arterial hypertension, including those with cardiovascular events.
In addition, there is an inspiring possibility of using certain brain regions for targeted bioelectrical and magnetic stimulation to optimize neuronal activity related to cognitive function. We propose that our results will help to identify such targeted brain regions based on the spatial distribution of ERPs in individuals with certain types of cognitive impairment in individuals with arterial hypertension. We have indicated this in the Introduction section with literature reference (doi: 10.3389/fneur.2021.799058).
Reviewer 1. The study group is so small that the statistics are unreliable… .
Response. The study was declared to be non-randomized with a control group. Therefore, we were entitled to obtain reliable results using nonparametric statistics methods. We selected the groups as carefully as possible so that they were comparable in terms of the distribution of age, gender, medical history, and treatment received. The use of non-parametric criteria (Mann-Whitney, Kruskal-Wallis) makes it possible to obtain reliable results in small groups. We indicated at the end of the Discussion section the limitation of the study, that we studied a limited number of subjects.
Reviewer 1. …and the evaluation of cognitive function with the MoCa test is nonexistent.
Response. If we understood the Reviewer 1 correctly, the assessment of cognitive functions using the MoCa test is not relevant.
Our study specified the NINDS-AIREN criteria, which were used to diagnose vascular dementia in the main groups individuals (Memory Group and VS/E Group). The MoCA test allows specifying the leading cognitive disorders, which was studied in our article. There are reports in the literature on the use of subscales of the MoCA test to assess cognitive function in older adults (DOI: 10.1002/gps.5269), including in randomized controlled trial (DOI: 10.3389/fnagi.2017.00379).
Bottiroli S. et al. (2017) used the MoCA test subscales (executive functions, working memory, visual-spatial processing, language and orientation) as effectiveness criteria for cognitive skills training in elderly people in the 3D virtual reality platform Smart Aging [Bottiroli S.]. The use of data from the MoCA subscales allowed the authors to conclude that this virtual platform appears to be a good tool for assessing primarily executive functions, working memory and visual-spatial processes [Bottiroli S. doi: 10.3389/fnagi.2017.00379]. A double-blind randomized controlled trial of multidomain attention training for alertness, sustained attention, and visual-spatial attention in older adults with mild cognitive impairment (MCI) was performed [Yang et al. 2020]. Some of the indicators of the effectiveness of such training were the subscales of the MoCA test (sustained attention, and visual-spatial attention). Significant improvements in attention, memory, and orientation on the values of the MMSE and MoCA subscales and better results compared to the control group 6 months after performing multidomain cognitive training have been shown [Yang et al. 2020, doi: 10.1002/gps.5269].
These facts gave us reason to use data from the MoCA test subscales as a selection of the leading types of cognitive impairment (memory impairment and visual/spatial and executive impairment). At the end of the Discussion section, we pointed to these research studies.
We suggest that our findings on the spatial cerebral distribution of ERPs in individuals with various leading cognitive impairments (in our case, memory and visual/spatial and executive functions) by the MoCA test can be used as clarifying neurophysiological biomarkers for selecting patients for various types of cognitive training in older hypertensive adults.
But we also further indicated that in order to further develop the topic of the findings, special separate neuropsychological tests should be conducted in order to more precisely determine the character of memory disorders and visual-constructive impairments.
In general, we hope for understanding of our point of view; we are ready for a positive and effective dialogue on the topic of this article.

Reviewer 2 Report
The study is interesting and treating a relevant matter, however such points need to addressed in my point of view:
Introduction: it is a bit confusing the continuous shift between P300 and N1,P2,N2. In fcat, early components (P1, N1, P2 and N2 components) are presumed to reflect neuronal sensory processing and categorization, while later components (including P300 with P3a and P3b components) are believed to reflect cognitive attention and executive control processes (Woodman, G. F. (2010). A brief introduction to the use of event-related potentials in studies of perception and attention. Attention, Perception, & Psychophysics, 72, 2031-2046.). Such cortical potentials have a very different meaning and in my opinion it should be more differentiated, separated and described in the text the different state of art concerning all these potentials with respect to the objective of the study.
Line 36-40: there is the need for a reference supporting such statement.
Line 64: " the latency of the N2 component of ERPs in certain cortical areas of the brain that is 64 an early biomarker of cognitive impairment" - which cortical areas?
Line 69-71: there is the need for a reference supporting such statement.
Line 99 and Line 106 the bullet point need to be corrected, there should appear respectively "B" and "C"
Line 146-148: were the different kinds of medicines omogeneously distributed among the three experimental groups (Control Group, VS/E Group and Memory group)? Otherwise the results could be affected by such bias.
Line 150-153: the same just above question also for complementary therapy
Line 249: M Group stands for Memory Group? In some parts of the article it is reported as Memory Group, in other M Group, please employ only one way for referring to that experimental group
Figure 1 caption: "Fragments of the spatial distribution of ERP P300 parameters" AND N2 component: in my opinion authors should mention also N2 component even in the initial descritpion of the Figure.
Discussion/conclusion: in my opinion the discussion about visuo-spatial attention should discuss the fact that further investigation employing visual evoked potentials would provide relevant data concerning such matter
Line 391: "Hypernensive" is a typo
Author Response
We are very grateful to the Reviewer 2 for the great work and attentive attitude to our scientific work. We tried to take into account all the comments and suggestions and made changes to the text of our article. It definitely made our research work better.
Reviewer 2. Introduction: it is a bit confusing the continuous shift between P300 and N1,P2,N2. In fact, early components (P1, N1, P2 and N2 components) are presumed to reflect neuronal sensory processing and categorization, while later components (including P300 with P3a and P3b components) are believed to reflect cognitive attention and executive control processes (Woodman, G. F. (2010). A brief introduction to the use of event-related potentials in studies of perception and attention. Attention, Perception, & Psychophysics, 72, 2031-2046.). Such cortical potentials have a very different meaning and in my opinion it should be more differentiated, separated and described in the text the different state of art concerning all these potentials with respect to the objective of the study.
Response. We agree with the Reviewer 2 that the presentation of data on the ERPs components in the text was not always accurate and consistent. We tried to differentiate the values of the N2 and P300 components in the text. We presented the main neurophysiological sense of N2 not as a sensory component (which differs from the components of P1, N1), but as a reflection of the recognition of significant stimuli and cognitive control with reference to authoritative literary sources. We have added sentences to the Introduction section to clarify the meaning of all the ERPs components and references.
The early ERPs components (P1 N1) are related to sensory perception [Woodman, doi: 10.3758/APP.72.8.2031], the N2 component is related to the process of meaningful stimulus recognition and cognitive control, and the N2-P3 (or P300) component reflects the motor component of ERPs in decision making associated with recognizing a meaningful stimulus [Warren, et al. doi: 10.1097/WNR.0b013e32834bbe1f].
Reviewer 2. Line 36-40: there is the need for a reference supporting such statement
Response. The link to relevant literature source has been added.
- Ishikawa, H.; Shindo, A; Mizutani, A.; Tomimoto, H.; Lo, E.H.; Arai, K. A brief overview of a mouse model of cerebral hypoperfusion by bilateral carotid artery stenosis. J Cereb Blood Flow Metab. 2023,8,271678X231154597. doi: 10.1177/0271678X231154597
- Chang, W.E.; Chang, C.H. Vascular Cognitive Impairment and Dementia. Continuum (Minneap Minn). 2022,28(3),750-780. doi: 10.1212/CON.0000000000001124
Reviewer 2. Line 64: "the latency of the N2 component of ERPs in certain cortical areas of the brain that is 64 an early biomarker of cognitive impairment" - which cortical areas?
Response 2. We have added this sentence.
It has been shown that it is the prolonged latency of the N2 component ERPs in the central, frontal, parietal, and left temporal brain regions, but not the P300 component, that may be a biomarker of cognitive impairment in middle-aged individuals with increased blood pressure.
Reviewer 2. Line 69-71: there is the need for a reference supporting such statement.
Responce. We hypothesaze that prediction of cognitive impairment progression by ERPs data is more effective for certain leading types of impairment, particularly against a background of cardiac rehabilitation and optimization of cerebral blood flow in hypertensive patients. The basis for this assumption is the advances in the prognostic value of ERPs during transcranial magnetic stimulation of precisely selected cortical areas of the brain for diagnosing specific cognitive impairments, such as those associated with impaired visual and spatial functions after a stroke [doi: 10.3389/fneur.2021.799058]. We have added this sentence.
Reviewer 2. Line 99 and Line 106 the bullet point need to be corrected, there should appear respectively "B" and "C"
Response. We have corrected these mistakes
Reviewer 2. Line 146-148: were the different kinds of medicines omogeneously distributed among the three experimental groups (Control Group, VS/E Group and Memory group)? Otherwise the results could be affected by such bias.
Line 150-153: the same just above question also for complementary therapy
Response. In the Methods section, we entered information by groups indicating the number of people who received combination medication therapy.
Reviewer 2. Line 249: M Group stands for Memory Group? In some parts of the article it is reported as Memory Group, in other M Group, please employ only one way for referring to that experimental group
Response. We have corrected the names of the groups throughout the article and given them the unified names - Control Group, Memory Group and VS/E Group.
Reviewer 2. Figure 1 caption: "Fragments of the spatial distribution of ERP P300 parameters" AND N2 component: in my opinion authors should mention also N2 component even in the initial descritpion of the Figure.
Response. We have added to the title of the Figure: Fragments of the spatial distribution of ERP N2 and P300 components… .
Reviewer 2. Discussion/conclusion: in my opinion the discussion about visuo-spatial attention should discuss the fact that further investigation employing visual evoked potentials would provide relevant data concerning such matter.
Response. We agree with the reviewer and have inserted a sentence about this method at the end of the Discussion section.
Identification of visual cognitive evoked potentials will be useful in the study of leading visual-spatial disorders developing in hypertensive individuals. The combined study of the spatial distribution of evoked brain potentials with different afferent inputs (both auditory and visual) in persons with cardiovascular events (stroke, transient ischemic attack) may help to identify the most significant brain areas that can be used for targeted therapeutic intervention, for example, during transcanial magnetic stimulation.
Reviewer 2. "Hypernensive" is a typo
Response. We have corrected this error
Reviewer 2. Moderate English changes required
Response. We will perform a professional proofreading of the text after agreeing all comments and changes in the text with the Editorial Board.

Round 2
Reviewer 1 Report
The authors' answers did not convince me. I still stand by my comments, I believe that the work does not bring new knowledge to the topic.
Author Response
The novelty of our presented research consists in the evaluation of different prognostic significance of ERPs parameters in specific brain areas in elderly persons with different leading cognitive disorders (memory and visual and constructive impairments) against the background of arterial hypertension. At present, the prognostic value of neuropsychological tests for assessing the further development of vascular dementia, as opposed to dementia against a background of neurodegenerative disorders, remains difficult. This is especially true for "silent" strokes on the background of arterial hypertension (indicated by us with reference to Yang, T., 2015). The diagnostic value of ERPs in standard averaged EEG leads (frontal or median leads only) may also be insufficient for predicting cognitive impairment in patients with arterial hypertension, as demonstrated by our results, when leading memory impairment and other cognitive functions are relatively intact.
We have also shown for the first time that in individuals with leading visual-constructive impairments, assessed using the subscales of the MoCA test, a pronounced decrease in acoustic signal recognition time (N2) and motor response time (P300) in decision-making occurs in certain brain regions. This suggests combine (mixed) cognitive impairment (both vascular and neurodegenerative). We have accompanied this assumption in the Discussion of Results section with an appropriate reference (Papadaniil, C.D., 2016). For the first time, our results on the spatial brain distribution of ERPs with leading visual constructive disorders allow us to identify a group of individuals for further examination using specific neuropsychological and electrophysiological tests to clarify the types of both visual constructive disorders and in-depth examination for the detection of comorbid neurodegenerative diseases (primarily Alzheimer's disease). Separating such a group according to both the leading visual constructive impairments and the spatial pattern of ERPs parameter distributions will allow prescribing targeted therapy to these individuals using non-drug correction (including local transcranial magnetic stimulation), which we indicated in the Discussion section with references to the reference (Ye, LL et al., 2022).
Reviewer 2 Report
I am happy that authors found my previous review useful. However, I think that some issues still needs to be addressed.
Introduction: I do think that the description of the state of the art concerning the employment of ERPs in such clinical conditions should be expanded a little bit more.
Methods: when describing exclusion criteria for the main and the control group, it is not clear why authors did not report also corresponding items between groups (e.g. presence of hearing loss for the control group, but in the main group there is no mention of the employed threshold level of such index).
Concerning the distribution of the combination therapy (line 159-165): I am afraid it is still not clear the actual combination distribution of the therapies, because just performing the sum for instance of the control group persons treated with the different medical treatments, it turns out to be 19, whilst the number of participants in the control group was 13. Therefore, there should be a combination of drugs not clearly explained, maybe using a tabke for summarizing the possible drugs combinations with relative numbers of patients undertaking such medical treatment would be of help. The same for the complementary therapy.
Author Response
We are grateful to Reviewer 2 for suggestions to improve the text of our article.
Reviewer 2: Methods: when describing exclusion criteria for the main and the control group, it is not clear why authors did not report also corresponding items between groups (e.g. presence of hearing loss for the control group, but in the main group there is no mention of the employed threshold level of such index).
Response: We assumed that the exclusion criteria applied to all individuals in the sample.
We have clarified this fragment of the text:
Non-inclusion criteria for both control and main groups (n=46)
Reviewer 2: Concerning the distribution of the combination therapy (line 159-165): I am afraid it is still not clear the actual combination distribution of the therapies, because just performing the sum for instance of the control group persons treated with the different medical treatments, it turns out to be 19, whilst the number of participants in the control group was 13. Therefore, there should be a combination of drugs not clearly explained, maybe using a tabke for summarizing the possible drugs combinations with relative numbers of patients undertaking such medical treatment would be of help. The same for the complementary therapy.
Response: In the Methods section, we have inserted Table 1 with the distribution of types of therapeutic procedures by groups of people
Round 3
Reviewer 1 Report
The authors have done a good job of improving the manuscript according to the reviewers' comments. The current version seems to be clear and lucid. Despite the limitation of the number of patients (which was highlighted in the limitations), it seems that the work can contribute interesting knowledge in the field.
However, I would still suggest small additions before accepting the work:
1. it is worth standardizing the naming of ERP components - if we use wave descriptions as P300 then N2 should be analogously N200
2. in the discussion, it is worth mentioning the wide application of ERP not only in diseases involving damage to the nervous system but also in potentially non-neurological diseases that can affect cognitive function based on: https://pubmed.ncbi.nlm.nih.gov/33510336/ and https://pubmed.ncbi.nlm.nih.gov/26271272/
Author Response
We are grateful to Reviewer 1 for these comments and suggestions on our article.
Reviewer 1. It is worth standardizing the naming of ERP components - if we use wave descriptions as P300 then N2 should be analogously N200
Response. We have corrected N200 names in all text where possible.
Reviewer 1. In the discussion, it is worth mentioning the wide application of ERP not only in diseases involving damage to the nervous system but also in potentially non-neurological diseases that can affect cognitive function based on: https://pubmed.ncbi.nlm.nih.gov/33510336/ and https://pubmed.ncbi.nlm.nih.gov/26271272/
Response. We understand the importance of these researchers and have added about it at the end of the Discussions section, as well as added recommended literature references.
Comorbid autoimmune conditions such as autoimmune Hashimoto's thyroiditis [ ...] or Sjögren's disease [ ...] contribute to cerebral mechanisms of prolongation of significant stimulus recognition time (N200) and decision time (P300), especially in older persons. Although the present study did not include people with the above pathology in the sample, a more thorough examination of patients is needed in the future because systemic autoimmune diseases may also influence the results of ERPs.
Reviewer 2 Report
Dear Authors,
I feel the manuscript very improved since its initial submission. I recommend its acceptance in the present form
Author Response
We are very grateful to Reviewer 2 for our efforts to improve the article. Of course, the English text of the article will be professionally proofread after the acceptance of the final version of the article.